# Biological Properties and Health-Promoting Functions of Laminarin: A Comprehensive Review of Preclinical and Clinical Studies

**DOI:** 10.3390/md20120772

**Published:** 2022-12-10

**Authors:** Shanmugapriya Karuppusamy, Gaurav Rajauria, Stephen Fitzpatrick, Henry Lyons, Helena McMahon, James Curtin, Brijesh K. Tiwari, Colm O’Donnell

**Affiliations:** 1School of Biosystems and Food Engineering, University College Dublin, Belfield, D04 V1W8 Dublin, Ireland; 2Department of Biological and Pharmaceutical Sciences, Munster Technological University, Clash, V92 CX88 Tralee, Ireland; 3Circular Bioeconomy Research Group, Shannon Applied Biotechnology Centre, Munster Technological University, V92 CX88 Tralee, Ireland; 4Nutramara Ltd., Beechgrove House Strand Street, V92 FH0K Tralee, Ireland; 5School of Food Science and Environmental Health, College of Sciences and Health, Technological University Dublin, D01 K822 Dublin, Ireland; 6Teagasc Food Research Centre, Department of Food Chemistry and Technology, Ashtown, D15 KN3K Dublin, Ireland

**Keywords:** laminarin, bioactive compounds, algal polysaccharide, biological activity, preclinical, clinical, human health, biomedical application

## Abstract

Marine algal species comprise of a large portion of polysaccharides which have shown multifunctional properties and health benefits for treating and preventing human diseases. Laminarin, or β-glucan, a storage polysaccharide from brown algae, has been reported to have potential pharmacological properties such as antioxidant, anti-tumor, anti-coagulant, anticancer, immunomodulatory, anti-obesity, anti-diabetic, anti-inflammatory, wound healing, and neuroprotective potential. It has been widely investigated as a functional material in biomedical applications as it is biodegradable, biocompatible, and is low toxic substances. The reported preclinical and clinical studies demonstrate the potential of laminarin as natural alternative agents in biomedical and industrial applications such as nutraceuticals, pharmaceuticals, functional food, drug development/delivery, and cosmeceuticals. This review summarizes the biological activities of laminarin, including mechanisms of action, impacts on human health, and reported health benefits. Additionally, this review also provides an overview of recent advances and identifies gaps and opportunities for further research in this field. It further emphasizes the molecular characteristics and biological activities of laminarin in both preclinical and clinical settings for the prevention of the diseases and as potential therapeutic interventions.

## 1. Introduction

Algal derived polysaccharides have gained much interest owing to their abundance as the main cell wall constituent as well as their unique physicochemical and biological properties [1]. Marine algal polysaccharides, including laminarin, fucoidan, ulvan, carrageenan, and alginate, have been widely explored for food, pharmaceutical and biomedical applications [2,3]. Polysaccharides are building blocks of about 30–50 monosaccharide units linked together by glycosidic bonds, together with many complex sugars to form cross-linked high–molecular weight biological macromolecules [4]. Initially, polysaccharides were used as thickening agents in industrial applications but recently have attracted much attention from the scientific community for their reported therapeutic potential [2,5]. The demand for novel, pure and highly biologically active polysaccharides is rapidly increasing, and their increased interest from researchers in exploiting natural algal resources. Algal biomass is a renewable bioresource and the recovered polysaccharides are safe, biodegradable, and biocompatible [2].

Laminarin is a relatively underexploited water-soluble polysaccharide present in brown algae that has been reported to exhibit potential therapeutic properties. Laminarin is increasingly being explored for the development of functional food and nutraceuticals [6,7,8,9], as well as a functional material in biomedical applications due to its biodegradable, biocompatible, and non-toxic nature [10,11] (Figure 1). This bioactive compound possesses the therapeutic potential to enhance and promote health and may help protect against diseases, including cardiovascular diseases, metabolic disorders, cancer, diabetes, obesity, anti-inflammatory activity, osteoarthritis, oral diseases, multiple sclerosis, as well as Alzheimer’s and Parkinson’s diseases, and vision-improving agents [12,13,14,15,16,17]. Several studies concluded that laminarin is also an excellent source of dietary fibre and a modulator of intestinal metabolism. Laminarin acts as a modulator through its effect on intestinal pH and short-chain fatty acids (SCFAs) production and is reported to enhance health, improve immunity, and treat and prevent diseases [6,9,12,13,16,18].

Despite the reported benefits of laminarin, there are limited studies based on cell culture investigations or biomedical engineering applications, and no commercial biomedical products are available in the market [2]. A limited number of studies on preclinical and clinical trials of laminarin from different brown macroalgae are reported in the literature [3,4,5]. It is therefore timely to review the therapeutic potential, biomolecular functions, multifunctional bioactivities, and possible health applications of laminarin (Table 1). In this review, the molecular structure and mode of action of laminarin from brown macroalgae are reviewed using preclinical and clinical reported studies. This emphasizes the biological activities of laminarin reported in preclinical and clinical case studies for the prevention of diseases and potential therapeutic interventions. This review provides scientific knowledge to underpin future novel biomedical applications of laminarin.

## 2. Structure and Molecular Characteristics of Laminarin

Structurally, laminarin is a water-soluble branched polysaccharide that consists of (1–3)- β-d-glucan with β (1–6)-linkages/branching of 20–25 glucose units, depending on the habitat, harvesting season and location (Figure 2) [48,51]. Brown macroalgae are reported to have around 350 mg/g laminarin content (on a dry basis), mostly present in the fronds part, which is influenced by algal species, harvesting season, geographic location, habitat, population age and method of extraction [45]. Laminarin possesses an average molecular weight of approximately 5 kDa depending on the degrees of polymerization. Additionally, depending on the type of sugar, the two forms of chains, M chains (end with terminal 1-O-substituted D-mannitol residues) or G chains (end with glucose residues), were observed at the reducing end. Rajauria et al. [52] identified the molecular weight of the purified laminarin in the range of 5.7–6.2 kDa. In particular, laminarin possesses a lower molecular weight than other polysaccharides reported in seaweed. The low molecular weight of laminarin has antioxidant activity due to carbonyl groups, which can improve lipid peroxidation.

Laminarin has been extracted from different brown macroalgal species, including *Laminaria digitata*, *Saccharina latissima*, *Laminaria japonica*, *Ecklonia kurome*, and *Eisenia bicyclis,* and to a lesser extent in Ascophyllum, Fucus, and Undaria species obtained from Asian and European countries [3,21,53,54]. The biological activity of laminarin depends on the molecular size, extraction methods, type of sugar, type of linkage and molecular geometry. In addition, the structure and biological activities of laminarin are influenced by environmental factors. For instance, a decrease in nitrite and nitrate in water as a nitrogen source stimulates the synthesis of laminarin [26]. Laminarin solubilizes in aqueous media or organic solvents, and in cold or hot water. Therefore, different conventional extraction approaches such as grinding, precipitation in an acid or alkaline medium, ultrafiltration, and dialysis with different molecular weight cut-off membranes have been employed to extract laminarin [26,46,55]. Recently, emerging technologies, such as ultrasound-assisted, microwave-assisted, and hydrothermal-assisted extraction processes, have been employed for the fast and cleaner recovery of laminarin with enhanced bioactivity [45,52,53]. Some reports have suggested that specific chemical modifications or treatment with processing techniques could enhance the bioactivity of laminarin [56,57,58,59]. Various processes, e.g., irradiation, sulfation, reduction, and oxidation, have been investigated for the modification of laminarin structure to improve the physicochemical, biological, and mechanical properties. Sulfated laminarin was reported to possess antitumor activity in human colorectal adenocarcinoma cells [57], prostate cancer cells, human melanoma SK-MEL-28, and colon cancer DLD-1 cells [21], anticoagulant activity [58], degradation by gamma irradiation [59], inhibition of cell proliferation through the activation via both specific receptor-mediated and mitochondria-mediated apoptotic pathways, and antimetastatic potential [60,61].

## 3. Biological Effects of Laminarin

Many laminarin research studies [20,21,26,28] reported multifunctional biological properties for potential therapeutic applications. Chemical modification of the structure of algal polysaccharides can improve their solubility and biological properties, which facilitates their potential use in biomedical applications and clinical trials. While recent in vitro and in vivo studies highlighted the biological activities and functional and physicochemical properties of laminarin, further in-depth studies are still required to investigate mitochondrial pathways relevant to the inhibition of proliferation and induction of cell apoptosis [7,57]. While recent in vitro and in vivo studies highlighted the biological activities, and functional and physicochemical properties of laminarin, further in-depth studies are required to investigate mitochondrial pathways relevant to the inhibition of proliferation and induction of cell apoptosis [7,57]. The research studies on brown seaweed for various biological activities with respect to the diseases are listed in Table 2.

### 3.1. Antioxidant

Rajauria et al. [52] reported that the molecular weight of laminarin influences antioxidant potential. In particular, crude extracts of laminarin have higher activity than purified and commercial products. Rajauria et al. [69] reported that laminarin was screened for its potential antioxidant capacity and found to have a significant radical scavenging capacity against free radicals and metal ions. These results showed that macroalgae are a rich source of natural antioxidants [9,19,53,70] in food and cosmetics [11,21].

Garcia-Vaquero et al. [53] investigated the antioxidant activity in laminarin from *Laminaria digitata* using DPPH and FRAP methods. The antioxidant and antimicrobial activities of crude laminarin extract were also examined by Kadam et al. [22] who confirmed a higher inhibition rate in scavenging of free radicals as antioxidant potential. Choi et al. [59] confirmed the therapeutic potential due to the antioxidant property of laminarin and outlined the interconnection with anti-inflammatory potential and a role in the activation of an immune response. Liu et al. [71] reported antioxidant activities against oxidative damage caused by reactive oxygen species (ROS) and free radicals. Preclinical studies were conducted to examine the antioxidant potential of laminarin using an animal model. Cheng et al. [72] showed using experimental rat studies that laminarin is a pulmonary oxidation and lipid peroxidation, protective agent. Another study by Jiang et al. [73] conducted in porcine early-stage embryos demonstrated significantly increased intracellular glutathione levels, cleavage, hatching, and blastocyst formation rates by maintaining the mitochondrial function, up-regulating differentiation and pluripotency-related genes. In addition, antimicrobial activity was also observed against different microorganisms [54,58]. Recent research has focused on antioxidant activity for the development of functional food products [19,70].

### 3.2. β-Glucan Related Receptors

Laminarin has been widely studied and utilized for various applications because of its molecular interaction with the glucan-specific pattern recognition receptor, Dectin-1 [74]. However, in recent years, laminarin has been used as a ligand for pattern recognition receptors and modulates innate immunity as immunoregulatory potential. Laminarin sulfate mimics the effects on smooth muscle cell proliferation and basic fibroblast growth factor-receptor binding and mitogenic activity [61]. Specifically, C-type lectin receptor (CLR) Dectin-1 acts as a receptor responsible for binding fungal β-glucans and eliciting innate immune responses [75]. Laminarin stimulates antitumor and antimicrobial activity by binding to receptors, such as complement receptor 3 and β-glucan receptor, as well as dectin-1 on macrophages and white blood cells, which provides new insights into the innate immune recognition of β-glucans [59,76]. In certain cases, variants of β-glucans polysaccharides downregulated autoimmune inflammation and can be mediated by cell surface receptors [77].

### 3.3. Immunomodulatory

The chemical characterization and quantification of laminarin to evaluate biochemical and ecological potential are considered unique methods, based on the physicochemical properties [78]. In this aspect, hydrogen bonds are resistant to hydrolysis in the small intestine and laminarin is considered as a dietary fibre possessing immunomodulating and anticoagulant properties [68,79]. Laminarin can interact with a specific receptor of the immune system for biological properties and immunomodulatory action [23]. Laminarin enhances the immune system with a high accumulation of B cells and helper T cells [23]. Recent research suggested that laminarin is an immune stimulatory molecule for cancer immunotherapy applications [12,21,68,78]. In this aspect, protein exhibited host–pathogen interactions and immunomodulatory potential, whereas lipids including long-chain fatty acids and short-chain fatty acids improve the healthy metabolism in cardiovascular and obesity-related diseases [6]. In addition, vitamins can also trigger the metabolic pathways in human health [2,12,29,30,31,32,47]. Due to the properties of proteins, lipids, and vitamins, laminarin possesses immunostimulatory activity. Kalasariya et al. [24] reported that laminarin promotes the adhesion of human skin fibroblast cells for wound healing and human osteoblast cells for bone formation in in vitro studies, demonstrating its immunostimulatory properties. In addition, laminarin was shown to inhibit tumor growth and metastasis due to its immunostimulatory potential [21,30].

The therapeutic potential of laminarin for the treatment of diseases was investigated in clinical trials conducted in pigs. The study by Rattigan et al. [62] reported that laminarin has the potential to prevent pathogen inflammation and proliferation and to enhance the fatty acids in the colon. Sweeney et al. [80] studied the immune response in the intestinal health of chicks by measuring growth performance, small intestinal morphology and function, and immune response. Similar results were also confirmed for gastrointestinal tract (GIT) health [67,81] in pig studies which showed a complex interaction between host genetics, environmental factors, and the gut microbiome [82] for inflammatory bowel diseases (IBD), including ulcerative colitis (UC) and Crohn’s disease. In addition, Vigors et al. [64] showed that dietary supplementation of laminarin also improves nutrient digestion, volatile fatty acids, and the intestinal microbiota using 16S rRNA gene sequencing in post-weaned pigs. Ostrzenski et al. [83] demonstrated the safety and efficacy of laminarin for resectoscopic cervical trauma in a non-pregnant patient population. Zaharudin et al. [33] showed that laminarin possesses hyperglycaemic and glycaemic potential in healthy adults using three-way blinded cross over trials. In addition, clinical studies conducted on humans reported that laminarin helps in improving postprandial hyperglycaemic and appetite control in healthy and normal-weight adults.

The anti-inflammatory activity of laminarin was shown using the gene expression profiles of anti- and pro-inflammatory markers [28,80], lower secretion of inflammatory cells in liver tissue and inflammatory mediators due to the β-glucan on immune cells and dietary fibre properties [68]. In addition, anti-inflammatory activity can be easily increased by the release of significant inflammatory mediators in laminarin such as hydrogen peroxide, calcium, nitric oxide, monocyte chemotactic protein-1, vascular endothelial growth factor, leukaemia inhibitory factor and granulocyte-colony-stimulating factor with enhancing expression of signal transducer and activator of transcription 1 (STAT1), STAT3, c-Jun, c-Fos and cyclooxygenase-2 mRNA. Rattigan et al. [62] showed that laminarin possesses anti-inflammatory potential by preventing pathogenic proliferation. Thus, it can also enhance diarrhoeal scores, body weight loss, and clinical variables linked with dextran sodium sulfate in pigs. A similar result was supported by O’Shea et al. [32] with systematic inflammation of laminarin. According to recent studies, laminarin can be used as a therapeutic agent for anti-inflammatory and immunostimulatory activities.

### 3.4. Wound Healing

Laminarin is considered an anticancer, antioxidant and anti-skin cancer agent [24]. Kadam et al. [54] reported that laminarin has potential application in wound healing as an effective antimicrobial agent. Zargarzadeh et al. [3] showed that marine-derived polysaccharides are considered multifunctional supporting biomaterials that can stimulate the healing process owing to their physicochemical and biological potential. Kadam et al. [26] reported that laminarin also promotes cell adhesion of human skin fibroblast cells for wound healing and proliferation in osteoblast cells for bone formation. Laminarin from Alaria species has been reported to promote cell adhesion of human skin fibroblast cells and cell proliferation in human osteoblast cells [84]. These studies on laminarin support its potential use in clinical investigations in cancer treatment. Due to its non-toxic, hydrophilic, and biodegradable properties, laminarin has potential as a wound-healing agent in modern medicine.

Sellimi et al. [17] reported that laminarin improves wound contraction, accelerates re-epithelization, collagen deposition, reconstitution of the skin tissue, and increases fibroblast and vascular densities in rats for wound healing effects. In addition, the beneficial role of laminarin promotes the healing process and skin regeneration. However, laminarin also has antibacterial and antioxidant properties which protect against free radical-mediated oxidative damage effect. Laminarin showed anticoagulant activity after structural modifications, such as sulphation, reduction, or oxidation. Laminarin was reported to be an effective polysaccharide in the prevention and treatment of cerebrovascular diseases due to its anticoagulant activity [3,6,34]. Kadam et al. [26] confirmed that laminarin possesses anti-coagulant activity due to its diverse biological properties. In addition, laminarin sulphate has been demonstrated in preclinical and clinical applications for anticoagulant activity, wound healing, angiogenesis, atherosclerosis, and cerebrovascular diseases.

Zargarzadeh et al. [3] noted that laminarin has been used in tissue engineering with cell function, which leads to tissue regeneration. The reduction of cholesterol levels by the active role of laminarin thereby lowers systolic blood pressure and levels of total cholesterol, free cholesterol, triglyceride, and phospholipid in the liver [26,35]. The clinical studies on laminarin extracted from marine brown seaweeds with respect to potential therapeutic applications are listed in Table 3.

### 3.5. Obesity

The tremendous influence of the human gut microbiota on health and intestinal disease is reported in recent studies [15,17,86,87]. Therefore, scientists can target gut microbiota-mediated immune systems for the treatment of cancer, diabetes, obesity, and cardiovascular diseases [86,87]. Earlier findings suggested that dietary fibre has a prebiotic effect because it supplies carbon to the fermentation pathways in the colon, thus supporting digestive health in humans and animals [36]. Brown seaweeds are a rich source of dietary fibre with an estimated content of 25–70% DW [79]. In particular, laminarin is highly resistant to hydrolysis in the upper gastrointestinal tract (GIT) because it is easily stabilized with its complex structures by inter-chain hydrogen bonds, and hence is considered as a good dietary fibre. Laminarin has the ability to stimulate and enhance the immune system in humans and is considered a biological response modifier [88]. In particular, laminarin from Laminaria digitata was shown to alter the gut microbiota composition through metagenomic compositional analysis and short-chain fatty acid (SCFA) analysis [15,17].

Zou et al. [81] demonstrated anti-obesity effects in pigs. A GIT pig clinical model was selected due to its similarity both morphologically and physiologically to humans. Many anti-obesity studies were carried out using laminarin in pigs [31,62,63,64,89], mice [25] and rats [68] studies. These studies confirmed that laminarin could stabilize diet-induced obesity and improve intestinal health, prevent post-weaning intestinal dysfunction, and maintain glucose homeostasis. Strain et al. [82] stated that laminarin can be considered a dietary supplement and anti-obesity functional food. Laminarin reduces the adverse effects of a high-fat diet by shifting gut microbiota towards higher energy metabolism as confirmed by 16S rRNA gene (V4) amplicon sequencing. Similar improvements in gut health were observed by Smith et al. [65], O’Doherty et al. [66], and Lynch et al. [37].

Preclinical studies using in vitro and in vivo models investigated laminarin properties using metabolic network analysis and proactive manipulation of the human gut microbiota [82,90]. Various human clinical trials were conducted to study the anti-obesity effects of laminarin. Odunsi et al. [38] in a randomized, placebo-controlled study evaluated the anti-obesity effects of laminarin for weight loss and demonstrated its effects as a short-term weight loss treatment for obese and overweight patients. Déjean et al. [85] conducted clinical studies on adults and employed metabolic network analysis and proactive manipulation to demonstrate the potential anti-obesity effects of laminarin and the effects on human gut microbiota.

### 3.6. Diabetes

Studies by Calderwood et al. [34] confirmed that laminarin acts as medicinal food or bio-therapeutic for type 2 diabetes mellitus by modulating the signaling pathways. Laminarin is a bioactive compound which possesses anti-hyperglycemic activity, can inhibit the cholesterol levels in serum, lower systolic blood pressure levels and stimulate the immune system [50]. The antidiabetic potential of laminarin from marine brown algae should be explored in drug development and nutraceutical applications. Hyperglycaemia can be reduced by inhibiting the carbohydrate-hydrolysing enzymes, such as α-amylase and α-glucosidase, for targeting type 2 diabetes mellitus. In addition to the reported in vitro antidiabetic activity of laminarin, the in vivo antidiabetic potential was also evaluated using animal models to confirm it’s hypoglycaemic effect through the inhibitory action on α-glucosidase and α-amylase enzymes [33].

Gunathilaka et al. [41] studied the antidiabetic potential of laminarin isolated from brown macroalgae and proved the inhibitory activity of rat lens aldose reductase enzyme in the presence of active porphyrin derivatives. Marine brown algae possess antidiabetic potential using the inhibitory activity of DPP-4 enzymes in a dose-dependent manner which is involved in glucose metabolism. Zaharudin et al. [33] reported that laminarin improves postprandial glycaemic and appetite control in healthy and normal-weight adults through clinical studies. However additional in-depth clinical studies are required to further investigate the biological activity of laminarin.

### 3.7. Cancer

Remya et al. [91] reported that laminarin showed anti-tumor activity against retinoblastoma Y79 cells and enhanced the activation of an immune response. Induction of apoptosis was also marked by the percentage of cells arrested in the G2/M phase using flow cytometry analysis and was further confirmed by a DNA fragmentation study. Similar studies were reported in other cancer cells, such as human melanoma SK-MEL-28 & SK-MEL-5 and colon cancer DLD-1 cells [92]; human colon cancer (HT-29), NK92-MI cells, human breast cancer (T-47D) and human skin melanoma (SK-MEL-28) [24,37,39,93,94]; human colorectal adenocarcinoma (HCT 116), and breast adenocarcinoma (MDA-MB-231) cells [38,95].

Ji et al. [57] reported that the sulfated modification of laminarin structure using the chlorosulfonic acid-pyridine method enhances antitumor activity on LoVo cells. Moreover, laminarin was shown to inhibit LoVo human colon cancer cell proliferation and induce LoVo cell apoptosis through a mitochondrial pathway. A similar finding was reported by Kadam et al. [26]. Menshova et al. [21] reported that the high molecular weight laminarin (19–27 kDa) from *Eisenia bicyclis* inhibited the colony formation of human melanoma SK-MEL-28 and colon cancer DLD-1 cells. In addition, the increase in 1, 6-linked glucose residues and the decrease of the molecular weight improved anticancer effect in laminarin.

George et al. [29] reported on the role of beta-1, 3-glucan derived from brown algae, which they showed had antitumor potential in hepatocellular carcinoma, colon cancer, leukemia, and melanoma. Their conclusions were also supported by Bae et al. [14]. Sulphated modification of laminarin structure enhances antitumor activity [57]. Laminarin also suppresses the formation of risk factors including SCFAs, indole compounds and ammonia for human colon cancer [57]. According to recent studies, laminaria supported the antitumor activity in brown macroalgae, which enhanced the activation of immune responses, cytotoxicity, apoptotic cell death, cell cycle arrest, colony formation, proliferation, and migration [21,24,37,38,93].

## 4. Therapeutic Effects of Laminarin as a Bioactive Agent

### 4.1. Antioxidant Activity

Laminarin is a dual regulator of apoptosis and cell proliferation, similar to β-Glucan [96]. Marine-derived antioxidant polysaccharides have three distinct mechanisms, including scavenging the ROS, regulating the antioxidant system or oxidative stress-mediated signaling pathways [97] (Figure 3a). The mechanism of β-glucan can activate different signaling pathways to regulate tumor cell proliferation, cell apoptosis, and cell cycle arrest, which can involve antioxidant, antitumor, and anti-inflammatory and immunostimulatory properties. Laminarin possesses potent antioxidant activity [52,59,69]. The mechanisms of antioxidant action in β-glucan with various pathways are shown in Figure 3b.

### 4.2. Immunomodulatory Activity

Laminarin has been reported to exert an immunostimulatory activity by the gene expression involved in inflammation and immune response, thereby it can stimulate the release of inflammatory mediators (Figure 4a). The specific receptors for β-glucan on dendritic cells (dectin-1), as well as interactions with other receptors, by innate immune cells (e.g., Toll-like receptors, complement receptor-3) are involved in the immune response to act as suitable therapeutic agents. Many studies have reported that β-glucan has similar biological activities to laminarin. Similar to β-glucan, laminarin has been reported to have activated innate with adaptive immunity, which induces humoral and cell-mediated immune responses and stimulates the production of proinflammatory molecules, such as complement components, IL-1α/β, TNF-α, IL-2, IFN-γ, and eicosanoids, as well as IL-10, IL-4, and proliferation of monocytes and macrophages (Figure 4b) [74,98].

The anti-inflammatory mechanism of laminarin is to stimulate the induction of innate immunity with certain ligands such as β-glucan. In particular, the antigens present in the tumor environment can stimulate an immune response in tumor cells and inhibit inflammation. Many preclinical and clinical studies have shown that β-glucan can enhance the antimicrobial activity of inflammatory macrophages, monocytes, and neutrophils, resulting in the maturation of target cells and an increased proinflammatory cytokine and chemokine release, stimulation of adaptive immune cells, including CD4+ T cells, CD8+ cytotoxic T lymphocytes (CTL) and B cells through the secretion of pro-inflammatory cytokines by T cells. In other cases, it also causes cell apoptosis with the release of ROS in the tumor microenvironment resulting in the destruction of tumor cells from oxidative stress [75]. Therefore, laminarin is considered as an immune-modulator agent and can be used as a synergic treatment in cancer and other inflammatory-related diseases.

Laminarin possesses anti-coagulant and inflammatory activities due to the modulation of innate immunity in specific metabolic pathways [99]. Furthermore, laminarin can reduce systolic blood pressure, cholesterol absorption in the gut, as well as cholesterol and total lipid levels in both serum and liver (Figure 5a). Microfold cells (M cells) are involved in transportation of antigen and particles to immune cells for modulating adaptive and innate immune responses. The activation of innate immunity by β-glucan is initiated by its binding the specific β-glucan receptor dectin-1 on macrophages. Thereby, it increases the inflammatory activity, enters the bone marrow, and stimulates the production of immune cells (Figure 5b).

### 4.3. Wound Healing

β-glucan was shown to have potential as a complementary therapy to manage various skin diseases and conditions due to its pluripotent activity [77]. The various cellular and extracellular matrix components and cells (keratinocytes, fibroblasts, endothelial cells, mast cells, nerve cells and leucocyte subtypes) participate differently in the three overlapping phases (inflammation, cell proliferation and tissue remodeling) in the healing process. Due to the antibacterial activity of laminarin, it also acts as a suitable wound healing agent with great stability. It can be mediated primarily by cell surface receptors, including immunocytes and cutaneous cells. The indirect signaling pathway through various cytokines of macrophages and direct signaling pathway on keratinocytes and fibroblasts cells are the two modes of immunostimulatory mechanisms enhancing wound healing. At the same time, growth factors from activated macrophages support cellular proliferation, angiogenesis, and reepithelialisation and an increase in wound tensile strength. Laminarin can accelerate the healing process in chronic and acute wounds by prolonging the inflammatory phase. The macrophages in granulation tissue were stimulated by β-glucan. Therefore, it acts as a source of growth factors and inflammatory cytokines (IL-6, IL-1, and TNFα), this pro-inflammatory event being mediated by the Dectin-1 receptor. In recent research, algal polysaccharides were investigated using green synthesis technology for their healing effects [100].

### 4.4. Anti-Obesity Activity

Laminarin significantly decreases high-fat diet-induced body weight gain and fat deposition. It also reduces blood glucose level and glucose tolerance. It enhances serum glucagon-like peptide-1 (GLP-1) content and the mRNA expression level of proglucagon and prohormone convertase 1 in the ileum. The mechanisms involved in anti-obesity activity have potential to be used to treat obesity and to maintain glucose homeostasis. β-Glucan is predominantly present in the cells in the cell walls of cereals, yeast, bacteria, and fungi, with significantly differing physicochemical properties [101]. However, laminarin is a type of β-glucan isolated from brown seaweed composed of D-glucose with β-(1, 3) linkages (Figure 6). However, laminarin promoted GLP-1 secretion and c-Fos protein expression dose-dependently. The expression of c-Fos protein is activated and regulated by mitogen-activated protein kinases (MAPKs) and the protein kinase C (PKC) signaling pathways [102]. Furthermore, glucose homeostasis and insulin sensitivity were improved. Laminarin promotes GLP-1 content via increasing intracellular calcium in enteroendocrine cells. Therefore, the long-term effects of laminarin counteract diet-induced obesity and improve glucose homeostasis by GLP-1 secretion [31].

### 4.5. Anti-Diabetic Activity

Hyperglycaemia can be reduced by inhibiting carbohydrate-hydrolysing enzymes such as α-amylase and α-glucosidase into glucose subunits. The inhibition of enzymes including aldose reductase, angiotensin-converting enzymes, dipeptidyl peptidase-4, and protein tyrosine phosphatase 1B can be exploited for the development of a therapeutic strategy [41,103]. In particular, dipeptidyl peptidase-4 increases GLP-1 levels to maintain hyperglycaemic conditions in patients with type 2 diabetes mellitus as compared with other pigments from microalgae and cyanobacteria as natural anti-hyperglycemic agents [31,104] (Figure 7a). Therefore, it acts as a therapeutic target for the development of antidiabetic drugs and different signaling pathways (Figure 7b). Recent studies reported that laminarin also causes inhibitory action involved in the antidiabetic mechanisms [31,43,44,105].

### 4.6. Antitumor Activity

Antitumor activity can inhibit cell apoptosis through a signaling pathway such as the mitochondrial pathway. Laminarin and laminarin sulfate possess direct cytotoxic activity against different types of tumor cells and have been shown to directly destroy cancer cells in studies using in vitro and in vivo models. Notably, laminarin also enhances T and B cells, macrophages, NK cells, and other immune cells, which can stimulate different complementary pathways and characterized by secretion of higher levels of anti-inflammatory cytokines for the regulation of the immune system. Algal polysaccharides help in the induction of apoptosis, cell proliferation, tumor angiogenesis, regulation of immune function and improve the effects of chemotherapy drugs on tumor cells in cancer therapy [101,106]. Laminarin was shown to be a potential therapeutic agent for human colon cancer by inhibiting cell proliferation due to the sulfated modification changing the molecular structure and spatial conformation of the polysaccharide, leading to changes in biological activity. The hydroxyl group of a sugar unit can be replaced by a sulfate group in laminarin. Thus, the conformation of the sugar chains changes, and the formation of a non-covalent bond further adds to the advantages of laminarin for antitumor activity. The mechanism of antitumor activity for algal polysaccharides is outlined in Figure 8.

## 5. Conclusions and Future Prospects

Research on the antioxidant, anti-tumor, anti-coagulant, and immunomodulatory activities of laminarin have shown beneficial effects in preclinical and clinical studies. Recent laminarin research also supported other potential activities including anti-obesity, anti-diabetic, anti-inflammatory, wound healing and hepatoprotective functions which have not been adequately characterized to date. Many research studies have demonstrated the biological activities of laminarin using structural modification and molecular weight determination. Further in-depth studies are required to facilitate the development of functional food and nutraceuticals incorporating laminarin.

Preclinical research on laminarin using in vitro and in vivo studies is very limited. Further clinical studies are also required to examine signaling pathways to facilitate the development of therapeutics, functional foods, and nutraceuticals. Further research is required to investigate the potential of laminarin in GI health and gut microbiome applications.

Laminarin is an underexploited polysaccharide in brown macroalgae of the marine environment with strong potential in therapeutic and nutraceutical applications. This review summarizes the preclinical and clinical laminarin studies and highlights opportunities and research priorities to facilitate the development of new therapeutics, functional foods, and nutraceuticals.

## Figures and Tables

**Figure 1 marinedrugs-20-00772-f001:**
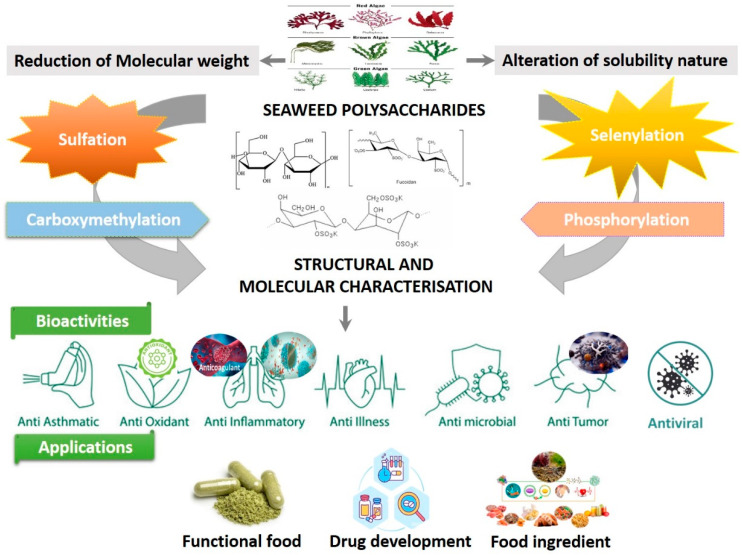
Seaweed polysaccharides have been investigated for commercial applications due to their reported properties and bioactivities.

**Figure 2 marinedrugs-20-00772-f002:**
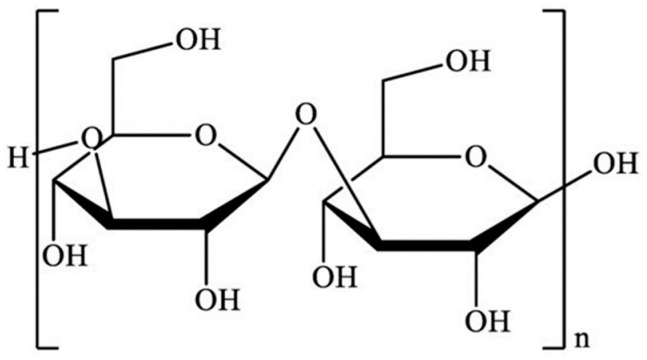
Structure of laminarin.

**Figure 3 marinedrugs-20-00772-f003:**
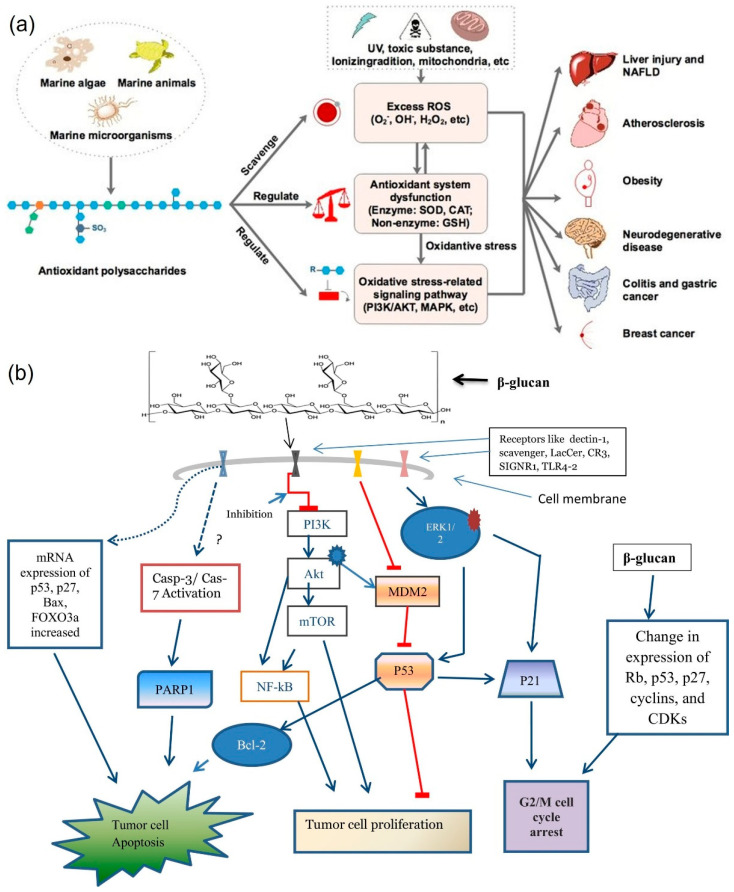
Antioxidant mechanisms. (**a**) Overview of marine-derived polysaccharides in alleviating oxidative stress-mediated diseases. Reprinted with permission from reference [97]. Copyright 2022, Zhong et al. (**b**) Antioxidant action of β-glucan. Reprinted with permission from reference [96]. Copyright 2022, Wani et al.

**Figure 4 marinedrugs-20-00772-f004:**
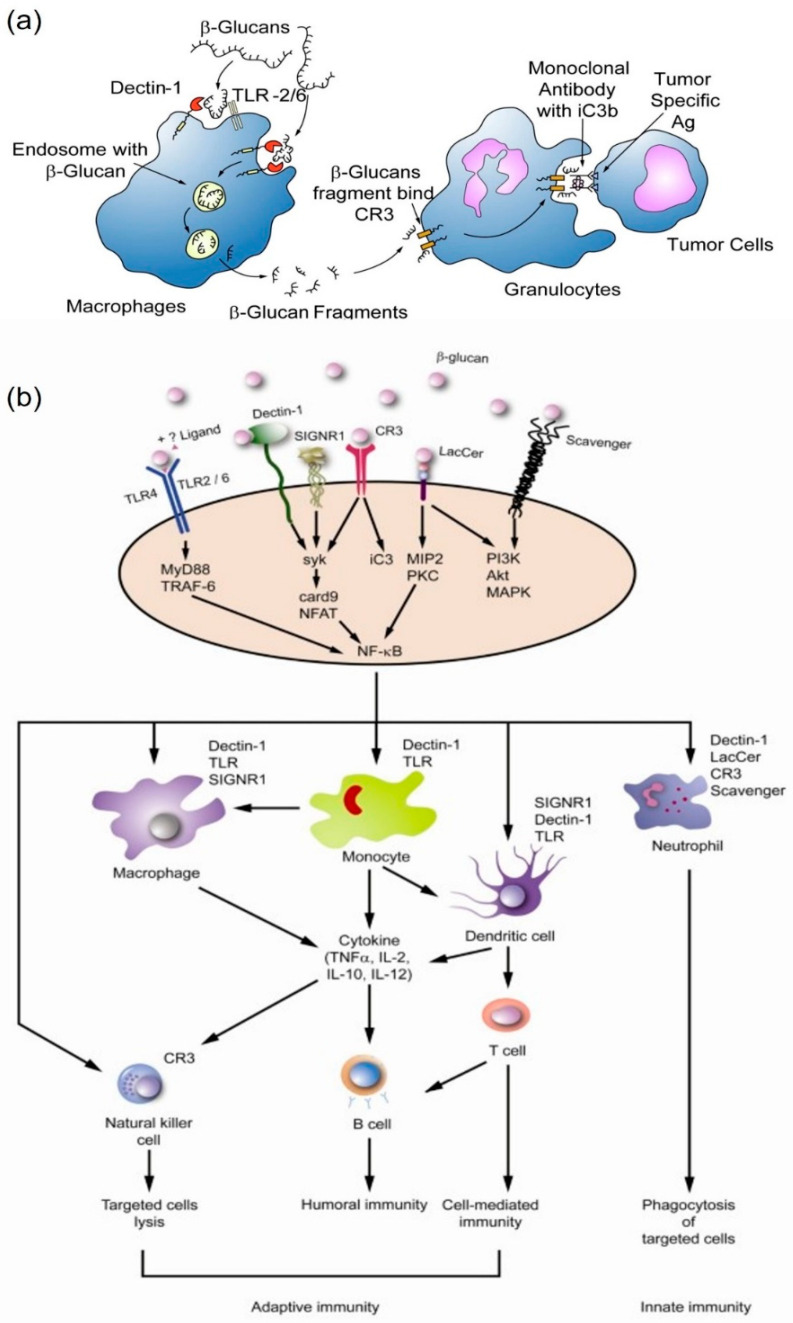
Immunomodulatory mechanism of β-glucan. (**a**) The uptake and subsequent actions of β-glucan on immune cells (**b**) Immune activation induced on a variety of membranes. Reprinted with permission from reference [74]. Copyright 2022, Chan et al.

**Figure 5 marinedrugs-20-00772-f005:**
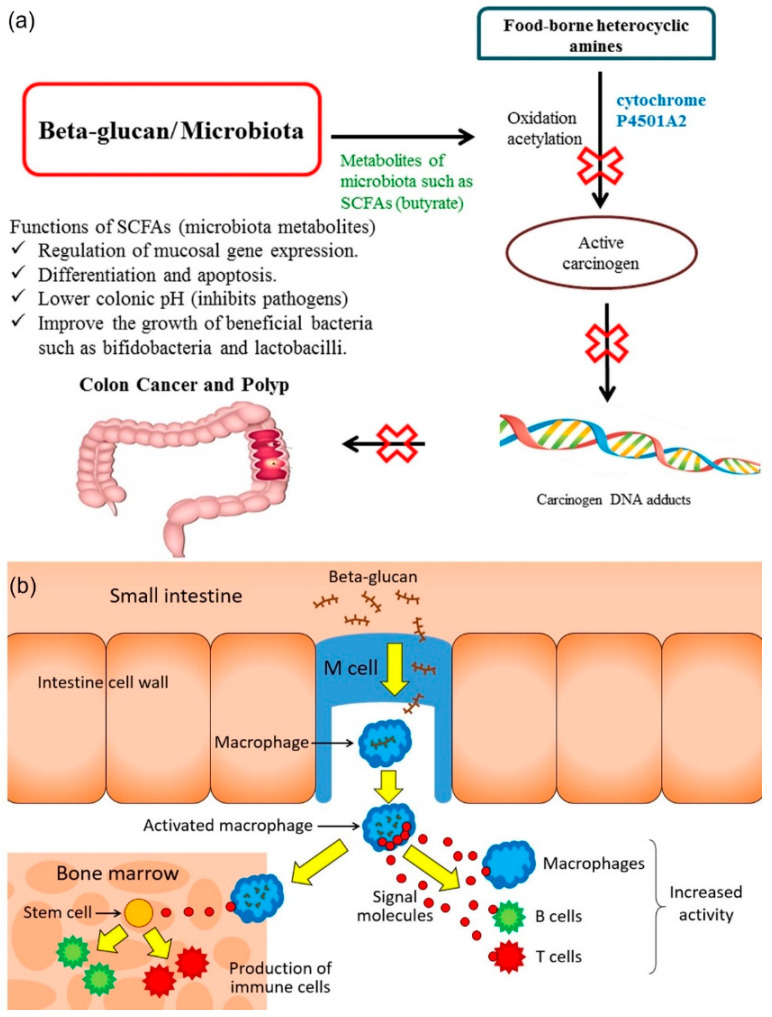
Mode of action in intestinal gut health (**a**) Action of absorption of β-glucan (**b**) influence of β-glucan in colon cancer via the gut microbiota. Reprinted with permission from reference [99]. Copyright 2022, Jayachandran et al.

**Figure 6 marinedrugs-20-00772-f006:**
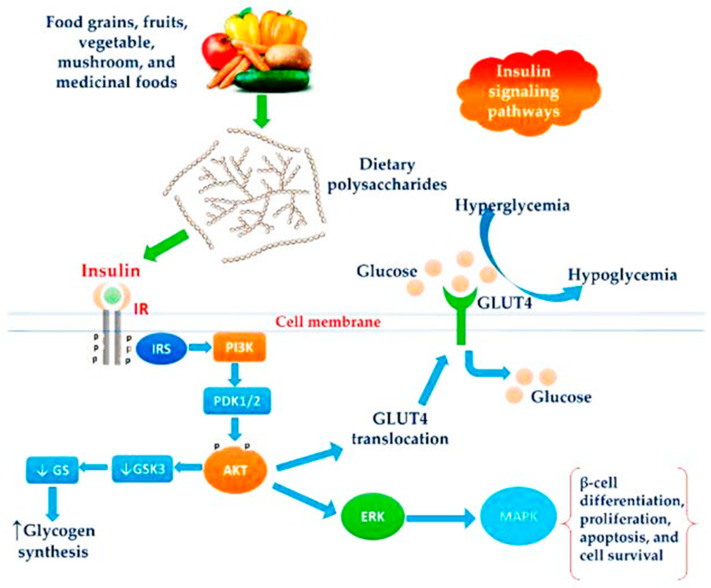
Schematic representation of the antiobesity action of dietary polysaccharides on the insulin signaling pathway. Reprinted with permission from reference [44]. Copyright 2022, Ganesan and Xu et al.

**Figure 7 marinedrugs-20-00772-f007:**
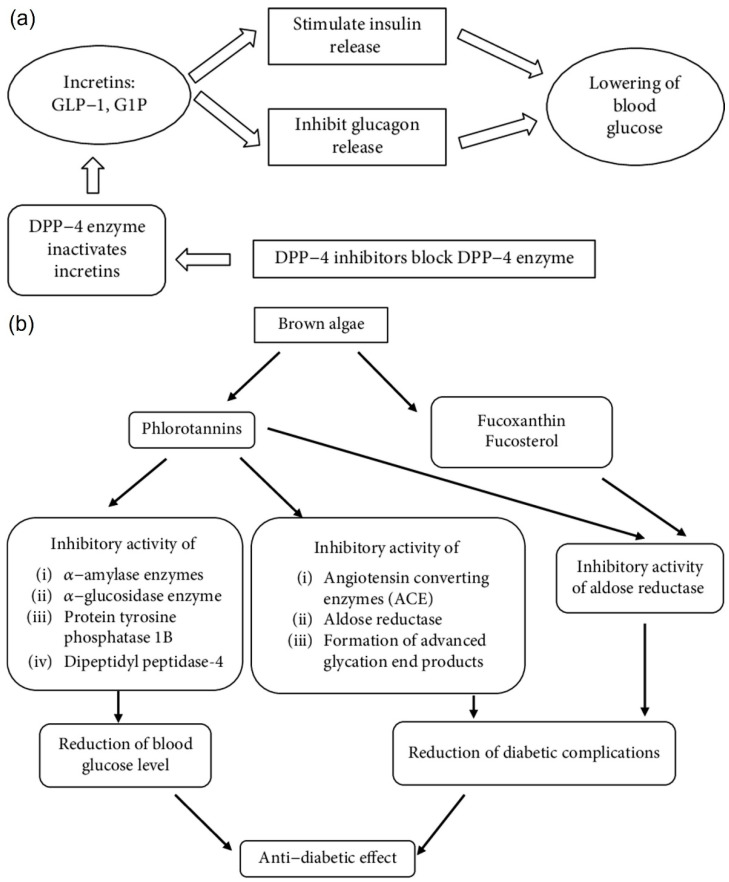
Antidiabetic mechanisms (**a**) Action of DPP-4 inhibitors on glucose homeostasis (**b**) Different antidiabetic mechanisms of active agents of brown algae. Reprinted with permission from reference [41]. Copyright 2022, Gunathilaka et al.

**Figure 8 marinedrugs-20-00772-f008:**
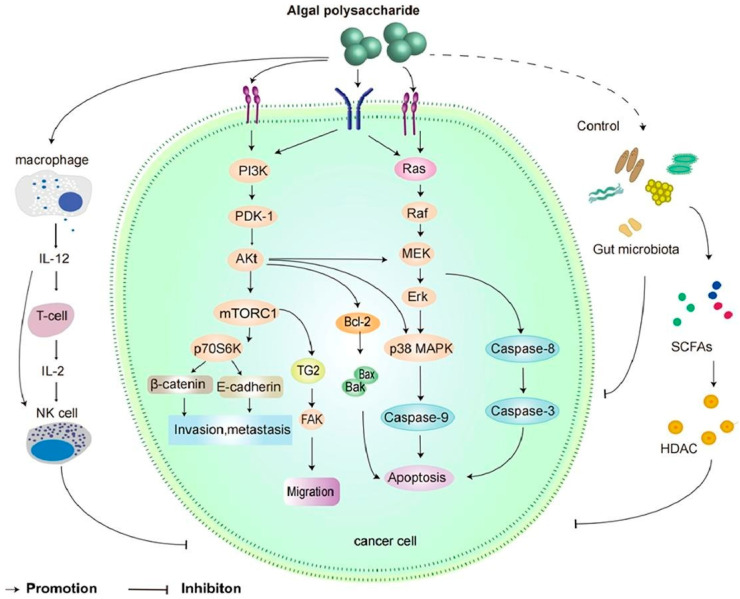
Anticancer mechanism of algal polysaccharide. 1: immunomodulation; 2: cytotoxicity; 3: cell cycle arrest; 4: NO-dependent pathway; 5: mitochondrial disruption. Reprinted from with permission from reference [101]. Copyright 2022, Ouyang et al.

**Table 1 marinedrugs-20-00772-t001:** Bioactive algal polysaccharides and their reported biological activities with therapeutic and nutraceutical potential.

Bioactive Algal Polysaccharides and Their Sources with Yield	Bioactivity Potential	Therapeutic and Nutraceutical Applications	Ref.
Laminarin *Laminaria digitata* (51.8%) *Saccharina latissima* (19 ± 2.6%) *S. japonica* *Alaria angusta* *Undaria pinnatifida* (3.2 ± 0.9%) *Dictyopteris delicatula* *Dictyota menstrualis* *L. saccharina* *Ecklonia cava* *Ascophyllum nodosum* *Sargassum* spp. (13.47%)	Antitumor Antimicrobial Wound healing Immunomodulatory Antioxidant activity Antibacterial and antifungal Modulatory effects on skin cells Hydro-gelling properties Anti-inflammatory Chemoprotective Neuroprotective Antiviral Antiallergic Antipruritic Hepatoprotective Anti-cholesterol Antidiabetic	Active ingredients Photoprotection Antiaging products Antioxidant Skin-benefiting activities Growth regulation Lipolytic activity Tissue engineering Cancer therapies Biofuel production Food industry Antimicrobial Health care and cosmetic uses Boosts immunity Prevents inflammation Biofertiliser	[3,10] [14] [15,19] [7,17,20] [11,21,22,23] [24,25] [26,27] [28] [13,22] [29] [30,31,32] [33] [34,35,36] [37] [38] [27,39,40] [41,42] [43,44]
Fucoidan *S. japonica* (1.26%), *A. angusta*, *U. pinnatifida* (1.5 ± 0.3%), *Dictyopteris* spp., *L. saccharina* *E. cava*, *A. nodosum*, *Cladosiphon okamuranus*, *Fucus vesiculosus* (18.22%), *L. japonica*, *F. evanescens* (4.44% (4.7%, *CaCl_2_;* 3.02% (5.11%, Ethanol)) *Dictyota menstrualis*, *S. polycystum, D. delicatula,* *Turbinaria conoides, S. latissimi,* *Spatoglossum asperum* *Cystoseira sedoides* *Coccophora langsdorfii*	Anti-tumor activity Anti-cancer activity Anti-viral activity Immunomodulatory activity Wound healing activity Anti-angiogenic activity Anti-allergy activity Anticoagulant activity Anti-diabetic activity Anti-hypolipidemic Anti-hyperglycemic activity Antioxidant activity Cognitive protective activity Anti-angiogenic activity Antimicrobial activity Anti-obesity activity Anti-inflammatory activity Anticancer	Immunomodulatory effect Tumor destruction Improved body weight and fasting blood glucose Anti-proliferative Antimetastatic Hypotriglyceridemic effects Anti-inflammatory response Neuroprotective effects Increase nitric oxide production. Activation of PI3K/Akt/eNOS-dependent pathways. Anti-inflammatory Anticancer	[10,35] [4,5] [5,24] [5] [7,19,36] [13] [25,45,46] [24] [41,42,47] [6] [5] [5,20] [8,23] [7] [22] [45] [6,10] [5]
Alginate *L. digitata* (51.8%), *A. nodosum*, *L. hyperborea*, *Macrocystis pyrifera*, *S. muticum* (13.47%) *S. binderi* (54%) *U. pinnatifida* (23.6 ± 1.2%)	Anti-obesity activity Antiviral Activity Anticancer Activities Antidiabetic Activities Antioxidant Activity Anti-inflammatory Anti-microbial Anti-coagulant Anti-ageing Anti-obesity	Overweight and Obesity Inhibition of α-amylase, pancreatic lipase and pepsin. Applications such as feed stabilizer, paper and welding rod coatings, and dye thickener in textile printing Drug delivery Pharmaceutical applications	[38] [2,24] [4,10,45] [18,47] [13] [48,45] [22,35] [42,49] [19,25]
Ulvan *Ulva pertusa* *Monostroma nitidum* and *U. pertusa* *U. pertusa* *Monostroma* spp. *M. nitidum* *Chlorella Pyrenoidosa* *U. intestinalis* *U. prolifera* *U. lactuca* *Ulva* and *Enteromorpha* spp.	Antioxidant activity Immunomodulating activity Antihyperlipidemic activity Antidiabetic activity Anticoagulant activity Anticancer activity Neuroprotective activity Immunostimulation Anti-inflammatory, Antiviral, Anticoagulant Gel-forming, skin protective and antioxidant properties	Influence plant immunity Triggering plant defense in several different plants Treatment of gastric ulcers Preventive/Therapautic Skin aging products	[7,20] [13] [35] [47] [19] [10] [12] [12] [24] [42] [36,25]
Carrageenan *Kappaphycus alvarezii,* and *Eucheuma dendiculatum* (50–55%) *Chondrus crispus,* *Sarcothalia crispata* *E. cottonii* (46.43%) *Furcellaria lumbricalis and* *Coccotylus truncates* (76.3%) *Mastocarpus stellatus* (28.65%)	Anticoagulant activity Antiviral activity Cholesterol-lowering activity Anti-tumor activity Immunomodulatory activity Antioxidant activity Anti-allergic activity	Enzymes hydrolyzing plant polysaccharides Food industry; mainly in dairy and meat applications Thickening, gelling, and stabilizing agent. Fertilizer Production of nanoengineered injectable hydrogels in tissue regeneration therapy	[7] [19] [49] [35] [10] [13] [18,20] [48,25]
Agar *Gracilaria lemaneiformis* (29.7 ± 1.9%) *G. vermiculophylla* (9.7–34.6%) *G. cornea* *Gelidium amansii*	Gelling and stabilizing properties Antioxidant Antidiabetic activity Neuroprotective and anti-neurodegenerative properties Scavenging properties, Antioxidant and gel-forming activity	Potential application in pharmaceutical industries Gelling agent Food application, including bakery, confectionery, dairy products, canned meat and fish products Preparation of bacteriological culture media Support for the three-dimensional cultures of human and animal cells Thickening agent	[7,48] [20] [47,35] [13] [18,42,49] [50,45] [36,25]

**Table 2 marinedrugs-20-00772-t002:** Summary of preclinical studies with respect to biological activity and health benefits of laminarin extract or fraction (*n* = 17).

Source	Extraction, Purification, Yield and Characterisation	Study Design	Model, Administration Dose, Route and Period	Biological Activities	Effects/Outcome of Laminarin Extract or Fraction and Related Products	Ref.
Edible seaweed species including *Laminaria digitata*	50 mL of boiling absolute ethanol (>99.5%) was added to 1.25 g of dried seaweed powder, then centrifuged after which the supernatant was removed. Ethanol extracts and water extracts of *L. digitata* were freeze dried. On experimental days, dried seaweed extracts were reconstituted in appropriate buffer for experimentation.	The in vivo anti-hyperglycemic activity.	C57/BL6 mice 500 mg/kg Orally 0–105 min	Hyperglycemic	Extracts of *L. digitata* strongly inhibited DPP-4 activity. Medicinal foods or biotherapeutics to tackle type 2 diabetes mellitus Targeted GLP-1 secretion, DPP-4 activity or alpha-glucosidase activity.	[34]
*Laminaria digitata*	The formulated agar in a brown seaweed, *L. digitata* was prepared. 10 g of processed seaweed sample was transferred to the beaker and mixed up with 100 mL of high-pure Milli-Q water then autoclaved at 121 °C for 1 h for extraction of water-soluble polysaccharides. The yield of formulated agar in a brown seaweed, *L. digitata* was found to be 40%. Characterized by FTIR, and SEM analysis.	The anti-skin cancer effect of formulated agar (FA) from *L. digitata* on dimethyl benzanthracene induced skin cancer mice.	Female Swiss albino mice 60–120 mg/mL Orally Four weeks	Anticancer Antioxidant Anti-skin cancer agent	FA from brown seaweed, *L. digitata* is a potent anti-skin cancer agent and antioxidant. Enhanced the antioxidant system	[29]
*Laminaria digitata*	Laminarin extracted from *L. digitata* (purity of 95%) was obtained from Sigma-Aldrich (St. Louis, MO, USA) and dissolved in sterile water before use. The response to laminarin treatment (0.1, 0.25, 0.5, 1, and 2 mg/mL) were used.	The anticancer effects of laminarin, a beta-1, 3-glucan derived from brown algae, have been reported in ovarian cancer (OC), colon cancer, leukemia, and melanoma.	A zebrafish xenograft model zebrafish embryos 0.5, 1, and 2 mg/mL Orally 24 h–72 h	Anticancer	Laminarin derived from brown algae had anticancer effects in human OC cells. Prevented tumor formation within the embryo yolks A novel OC suppressor.	[14]
*Laminaria* spp.	The laminarin-rich extract was obtained from Laminaria spp. using hydrothermal assisted extraction by applying optimized extraction conditions. The laminarin-rich extract was included in sufficient quantity to achieve a concentration of 200 ppm in the relevant treatment. The crude extract was partially purified in order to enhance the polysaccharide content by mixing the crude extract with pure ethanol followed by water and calcium chloride.	The effects of increasing dietary inclusion levels of laminarin in the first 14 d post-weaning on pig growth performance and weaning associated intestinal dysfunction.	Ninety-six healthy piglets 650 g/kg Lethal injection 28 days	Anti-obesity Gastrointestinal health	Laminarin-rich extract has potential to prevent pathogen proliferation. Enhanced the volatile fatty acid profile in the colon in a porcine model of colitis.	[62]
*Laminaria* spp.	The laminarin-rich extract was sourced from BioAtlantis Ltd (Tralee, Co. Kerry, Ireland). The extract was prepared by using water as an extraction solvent under optimum heating conditions. A single extraction was performed from Laminaria spp. to produce the extract. Appropriate quantity of laminarin as 100, 200 and 300 parts per million (ppm) was added.	The effects of dietary supplementation with laminarin on colonic health in pigs challenged with dextran sodium sulphate.	Forty-two healthy male pigs 200 ppm Orally 35 days	Anti-inflammatory	300 ppm of laminarin from a laminarin-rich extract as a dietary supplement, improved performance and prevented post-weaning intestinal dysfunction	[63]
*Laminaria* spp.	The laminarin enriched extract was sourced from BioAtlantis Ltd. The laminarin rich extract was obtained from Laminaria spp. using hydrothermal assisted extraction by applying optimised extraction conditions. The crude extract was partially purified in order to enhance polysaccharide content.	The effects of supplementing the diet of newly weaned pigs with 300 ppm of a laminarin rich extract, on animal performance, volatile fatty acids, and the intestinal microbiota using 16S rRNA gene sequencing	Fifty-four weaned pigs 300 ppm Orally 28 days	Anti-obesity Gut health	300 ppm of a laminarin rich macroalgal extract reduced post-weaning intestinal dysfunction in pigs. Promoted the proliferation of bacterial taxa Enhanced nutrient digestion by reducing the load of taxa	[64]
*Saccharina longicruris*	Laminarin (TCI Shanghai, China) was dissolved in saline to prepared stock solution of 50 mg/mL. One g/kg of laminarin by intragastric administration and characterized by western blot assay.	The effect of laminarin on energy homeostasis, mice were orally administrated with laminarin to test food intake, fat deposition, and glucose homeostasis.	Homeostasis model C57/BL6 mice 50 mg/mL Orally Four weeks	Anti-obesity	Laminarin counteracts diet-induced obesity. Maintained glucose homeostasis. Promoted GLP-1 secretion via increasing intracellular calcium in enteroendocrine cells.	[31]
*Laminaria digitata*	Laminarin derived from *L. digitata* was purchased from Invivogen. The endotoxin levels in the purified laminarin were evaluated using a Limulus gametocyte lysate (LAL) assay kit (Lonza, Gampel, Switzerland).	The effects of laminarin on the maturation of dendritic cells and on the in vivo activation of anti-cancer immunity using homozygous transgenic mice (OT-I and II).	C57BL/6 mice, OT-I and OT-II TCR transgenic mice, and C57BL/6-Ly5.1 (CD45.1) congenic mice 25 mg/kg Intrasplenic B16-OVA injection 24 h	Immunomodulatory	The purified laminarin stimulated and inhibited tumor growth and metastasis Potential immune stimulatory molecule for use in cancer immunotherapy.	[30]
*Laminaria* spp.	The crude seaweed extract was derived from Laminaria spp. (BioAtlantis Ltd.). The laminarin content of the supplements and the feed samples was determined by spectrophotometry using a commercial assay kit (Megazyme Ireland Ltd., Bray, Ireland).	The potent anti-inflammatory activities of the algal polysaccharides laminarin and fucoidan in the gastrointestinal tract of pigs	Dextran sodium sulfate-challenged porcine model Thirty-five pigs 300 mg/kg Oral administration 56 days	Anti-inflammatory	Laminarin has potent anti-inflammatory activities in the gastrointestinal tract. Reduced Enterobacteriaceae in proximal colon digesta. Decreased in colonic IL-6 mRNA abundance for its inflammatory effect	[32]
*Laminarin* spp.	A high-fat diet (HFD) and 1% laminarin-supplemented water (HFL). Laminarin (Sigma) supplementation was terminated at the fourth week, followed by continuation HFD for an additional 2 weeks.	The anti-obesity effects of the potential prebiotic, laminarin, on mice, fed a high-fat diet.	Laminarin-supplemented high-fat diet (HFL) mice model Eighteen female BALB/c mice 45% (*w*/*v*) Orally 6 weeks	Anti-obesity effects Gut Health and microbiota	Laminarin (Sigma) reduced the adverse effects of a high-fat diet by shifting gut microbiota towards higher energy metabolism. Laminarin could be used to develop anti-obesity functional foods	[25]
*Laminaria digitata, L. hyperborea* and *Saccharomyces cerevisiae*	Purified laminarin from *L. digitata* and *L. hyperborea* (990 g laminarin/kg) was sourced from Bioatlantis Limited and extracted. A basal diet supplemented with 250 parts per million (ppm) laminarin from *L. digitata.*	The effects of purified laminarin derived from *L. digitata, L. hyperborea* and *S. cerevisiae* on piglet performance in selected bacterial populations and intestinal volatile fatty acid (VFA) production	Thirty-two pigs 250 mg/kg Orally 28 days	Anti-inflammatory	Purified laminarin may be acting via a different mechanism from the insoluble β-glucan. Identified as natural biomolecules with immunomodulatory activity	[28]
*Laminaria digitata*	Purified laminarin (990 g/kg laminarin) was sourced from Bioatlantis Ltd., and extracted.A basal diet supplemented with 300, and 600 parts per million (ppm) laminarin from *L. digitata.*	The optimum inclusion level of laminarin derived from *L. digitata* on selected microbial populations, intestinal fermentation, cytokine and mucin gene expression in the porcine ileum and colon	Twenty-one Pigs 990 g/kg Orally 26 days	Gastrointestinal health	Dietary inclusion of 300 ppm purified laminarin appears to be the optimum dose. Reduced enterobacteriaceae populations Enhanced IL-6 and IL-8 cytokine expression in response to an ex-vivo LPS challenge	[65]
*Laminaria digitata*	Seaweed extracts derived from *L. digitata* were included at 2.8 g/kg. The extract contained both laminarin (112 g/kg), fucoidan (89 g/kg) and ash (799 g/kg) and was sourced from Bioatlantis Ltd.	The interactions between two different lactose (L) levels and seaweed extract containing laminarin and fucoidan derived from Laminaria spp. on growth performance, coefficient of total tract apparent digestibility (CTTAD) and faecal microbial populations in the weanling pig.	Two hundred and forty pigs 112 g/kg Orally 25 days	Intestinal digestibility	The inclusion of a high dietary concentration of laminarin increased the CTTAD of diet components and decreased the counts of *E. coli* in the faeces. Improved performance of pigs after weaning	[66]
*Laminaria hyperborea*	The seaweed extract was extracted from Laminaria spp. The seaweed extract contained laminarin (112 g kg^−1^), fucoidan (89 g kg^−1^) and ash (799 g kg^−1^) and was sourced from BioAtlantis Ltd.	The effect of dietary Laminaria-derived laminarin and fucoidan on nutrient digestibility, nitrogen utilisation, intestinal microflora and volatile fatty acid concentration in pigs	Thirty finishing boars/pigs 112 g/kg Orally 14 days	Anti-obesity effects Dietary and gut health	Reduced intestinal *Enterobacterium* spp. and increased in *Lactobacilli* spp. The dietary laminarin may provide a dietary means to improve gut health in pigs	[37]
*Laminaria* spp.	The seaweed extract was extracted from Laminaria spp. Seaweeds extracts of 1, 2 and 4 g/kg were used. The seaweed extract was sourced from BioAtlantis Ltd.	The interaction between different levels of lactose and seaweed extract derived from Laminaria spp. on growth performance and nutrient digestibility of weanling pigs.	384 piglets 1,2,4 g/kg Orally 21 days	Intestinal health	Pigs responded differently to the inclusion levels of seaweed extract at each level of lactose supplementation.	[67]
*Laminaria* spp.	Laminarin (purity of 90%). was provided by Goëmar (St Malo, France). The molecular weight was 4500 to 5500 g/mol and the purity was 90%. laminarin treated rats (LAM) received the same diet containing 5 g/100 g laminarin for 4 days followed by a dietary treatment of 10 g/100 g LAM for 21 days.	The hypothesis stated that LAM, a β (1–3) polysaccharide extracted from brown algae, can modulate the response to systemic inflammation.	Male Wistar rats 5 g/100 g for 4 days followed by 10 g/100 g Intraperitoneal 21 days	Hepatoprotective	The effects could be due to a direct effect of purified laminarin on immune cells, or to an indirect effect through their dietary fibre properties. Decreased serum monocytes number, TNF-α and nitrite.	[39]
*Laminaria saccharina*	Brown seaweeds, *L. saccharina* (Algoplus, Roscoff, France) were collected. Laminarin was extracted from ground algae (15–25 g) by addition of absolute ethanol, hot H_2_SO_4_, and hot HCl. The various methods of extraction of laminarin was implemented by partial characterisation.	The various methods of extraction of laminarin were compared by partial characterization and, on the other hand, to study the fate of this polysaccharide and its effects in the gastrointestinal tract in order to determine its potential as a dietary fibre in human nutrition.	Wistar rats 15–25 g (*w*/*v*) Orally 18 days.	Anti-obesity effects Gut health	Laminarin can be considered a dietary fibre No increased in the intestinal transit and stool output. Laminarin resisted hydrolysis in the human upper gastrointestinal tract.	[68]

**Table 3 marinedrugs-20-00772-t003:** Summary of clinical studies with respect to biological activity and health benefits of laminarin extract or fraction and related products (*n* = 4).

Source	Extraction, Purification, Yield and Characterisation	Study Design	Model, Administration Dose, Route and Period	Biological Activities	Effects/Outcome of Laminarin Extract or Fraction and Related Products	Ref.
*Laminaria* *digitata* and *Undaria pinnatifida*	The dried seaweed was soaked in 200 mL of water for 10 min, then rinsed and drained to remove excess water. Finally, they were cut into pieces and added with 0.5 g iodine enriched salt (6.5 mg iodine). The yield of dietary fibre (g/serving) about 1.8.	The effects of two brown edible seaweeds, *L. digitata* and *U. pinnatifida*, on postprandial glucose metabolism and appetite following a starch load in a human meal study as first trail.	A randomized, Three-way, blinded cross-over trial Twenty healthy participants 5 g (*w*/*v*) Orally 180 min	Hyperglycaemic	Concomitant ingestion of brown seaweeds may improve postprandial glycaemic and appetite control in healthy and normal weight adults, depending on the dose. Brown seaweeds inhibited the postprandial glucose and insulin response.	[33]
*Laminaria digitata* and *Eisenia bicyclis*	Laminarins from *L. digitata* and *Eisenia bicyclis* (Purity of 95%) were obtained from Sigma-Aldrich (St. Louis, MO, USA). Laminarin from *L. digitata* was reduced to laminaritol, to reduce background responses in the bicinchoninic acid (BCA) reducing sugar by enzyme kinetics assay. Purified by Crystallization, NMR, and electrophoresis and used for the human gut microbiota (HGM) study.	The symbiont Bacteroides uniform deploys a single, exemplar polysaccharide utilization locus (PUL) to access yeast β (1, 3)-glucan, brown seaweed β (1, 3)-glucan (laminarin), and cereal mixed-linkage β (1, 3)/β (1, 4)-glucan as first trial.	2441 adults	Anti-obesity effects Human gut microbiota	Fine-grained knowledge of PUL function Metabolic network analysis and proactive manipulation of the HGM. Purified Laminarin provided a validated set of molecular markers to identify β (1, 3)-glucan utilization potential among members of the HGM.	[85]
*Laminaria digitata*	The purified laminarin (LAM) from the brown algae *L. digitata* were prepared. The basal diet supplemented with 250 ppm purified LAM; basal diet supplemented with a seaweed extracts.	The effects of supplementing the diet with seaweed extracts on growth performance, small intestinal morphology and function, immune response and *Campylobacter jejuni* colonisation following an experimental challenge in young chicks as first trial.	Hundred- and thirty-five-day-old male Ross chicks 250 ppm Orally 13 days	Intestinal health	Supplementation with laminarin improved growth rate, positively modified small intestinal architecture and Impacted the intestinal immune response.	[80]
*Laminaria digitata*	A group of 30 patients were dilated preoperatively with Laminaria tents, prepared from *L. digitata*.	The safety and efficacy of *L. digitata* was examined to dilate the cervix before resectoscopic surgery as randomized trial.	Thirty patients Randomized trial	Resectoscopic cervical trauma	More than 8-mm cervical dilatation	[83]

## Data Availability

Not applicable.

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
