# Peer review of "Biological Properties and Health-Promoting Functions of Laminarin: A Comprehensive Review of Preclinical and Clinical Studies"

_marinedrugs, 2022, doi:10.3390/md20120772_

Round 1

Reviewer 1 Report

The results presented and reviewed here should be accompanied by data and compared adequately.Please improve the review by how much better laminarin responses are and values mg per g 

Please compare the studies with other studies which doesn't focus much on laminarin but maybe give a comparison of why Laminain is superior or where Laminarin is not much effective.

Prepare a comparative table to compare the activity of laminarin with other bioactive molecules

Please include the following references  

1. Paliwal, C., Nesamma, A. A., & Jutur, P. P. (2019). Industrial scope with high-value biomolecules from microalgae. In Sustainable Downstream Processing of Microalgae for Industrial Application (pp. 83-98). CRC Press.

2. Paliwal, C., Ghosh, T., Bhayani, K., Maurya, R., & Mishra, S. (2015). Antioxidant, anti-nephrolithe activities and in vitro digestibility studies of three different cyanobacterial pigment extracts. Marine drugs13(8), 5384-5401.

3. Ghosh, T., Bhayani, K., Paliwal, C., Maurya, R., Chokshi, K., Pancha, I., & Mishra, S. (2016). Cyanobacterial pigments as natural anti-hyperglycemic agents: an in vitro study. Frontiers in Marine Science3, 146.

4. Pradhan, B., Nayak, R., Patra, S., Bhuyan, P. P., Dash, S. R., Ki, J. S., ... & Jena, M. (2022). Cyanobacteria and Algae-Derived Bioactive Metabolites as Antiviral Agents: Evidence, Mode of Action, and Scope for Further Expansion; A Comprehensive Review in Light of the SARS-CoV-2 Outbreak. Antioxidants11(2), 354.

5. Younis, N. S., Mohamed, M. E., & El Semary, N. A. (2022). Green Synthesis of Silver Nanoparticles by the Cyanobacteria Synechocystis sp.: Characterization, Antimicrobial and Diabetic Wound-Healing Actions. Marine Drugs20(1), 56.

Author Response

RESPONSE TO REVIEWER 1

The authors appreciate the reviewer’s comments and have revised the text accordingly.

Comment-1: The results presented and reviewed here should be accompanied by data and compared adequately. Please improve the review by how much better laminarin responses are and values mg per g

Response:  Thank you for your suggestion. Data now modified and included in the revised manuscript.

Amended text is included in line number 82-85 as follows:

“~Brown macroalgae are reported to have around 350 mg/g laminarin content (on a dry basis), mostly present in the fronds part, which is influenced by algal species, harvesting season, geographic location, habitat, population age and method of extraction~”

Comment-2: Please compare the studies with other studies which doesn't focus much on laminarin but maybe give a comparison of why Laminain is superior or where Laminarin is not much effective.

Response:  Thank you for your suggestion. Text has been modified and included in the revised manuscript. This review paper is focused on the list of studies about laminarin and stated the importance of the laminarin and effective potential in the order of biological activities. See line 101, 123, 164, 246, 290.  We reported all studies that you mentioned here.

Comment-3: Prepare a comparative table to compare the activity of laminarin with other bioactive molecules

Response:  Thank you for your suggestion. We included this information in Table 1, line number 158-162. Text has been modified and included in the revised manuscript. 

Comment-4: Please include the following references 

  1. Paliwal, C., Nesamma, A. A., & Jutur, P. P. (2019). Industrial scope with high-value biomolecules from microalgae. In Sustainable Downstream Processing of Microalgae for Industrial Application (pp. 83-98). CRC Press.
  2. Paliwal, C., Ghosh, T., Bhayani, K., Maurya, R., & Mishra, S. (2015). Antioxidant, anti-nephrolithe activities and in vitro digestibility studies of three different cyanobacterial pigment extracts. Marine drugs, 13(8), 5384-5401.
  3. Ghosh, T., Bhayani, K., Paliwal, C., Maurya, R., Chokshi, K., Pancha, I., & Mishra, S. (2016). Cyanobacterial pigments as natural anti-hyperglycemic agents: an in vitro study. Frontiers in Marine Science, 3, 146.
  4. Pradhan, B., Nayak, R., Patra, S., Bhuyan, P. P., Dash, S. R., Ki, J. S., ... & Jena, M. (2022). Cyanobacteria and Algae-Derived Bioactive Metabolites as Antiviral Agents: Evidence, Mode of Action, and Scope for Further Expansion; A Comprehensive Review in Light of the SARS-CoV-2 Outbreak. Antioxidants, 11(2), 354.
  5. Younis, N. S., Mohamed, M. E., & El Semary, N. A. (2022). Green Synthesis of Silver Nanoparticles by the Cyanobacteria Synechocystis sp.: Characterization, Antimicrobial and Diabetic Wound-Healing Actions. Marine Drugs, 20(1), 56.

Response:  Thank you for your suggestion. We have included three of these references in line number 771, 777, and 780 of the revised manuscript.

Sentences are included in line number 444 and 472 as follows:

“~IL-1 and TNFα), this pro-inflammatory event being mediated by the Dectin-1 receptor. In recent research, algal polysaccharides were investigated using green synthesis technology for healing effects [107]. ~”

“~strategy [79, 95]. In particular, dipeptidyl peptidase-4 increases GLP-1 levels to maintain hyperglycaemic conditions in patients with type 2 diabetes mellitus as compared with other pigments from microalgae and cyanobacteria as natural anti-hyperglycemic agents [51, 102] (Figure 7a). Therefore, it acts as a therapeutic target for the development of anti-diabetic~”

Reviewer 2 Report

Biological properties and health-promoting functions of Laminarin: A comprehensive review of preclinical and clinical studies by Shanmugapriya Karuppusamy et al described the diverse pharmacological properties of laminarin, such as antioxidant, immunomodulatory and antiobesity. Moreover, mechanisms of action were summarized detailedly in this review. This is a well-organized comprehensive review, providing an overview of recent advances, and identifies gaps and opportunities for further research in this field.

Only minor revisions as following:

1. The structure in Figure 2 was too large.

2. P3L97: Italics for the species and genera names for ‘Laminaria digitata, Saccharina latissima, Laminaria japonica, Ecklonia kurome, Eisenia bicyclis, Ascophyllum, Fucus, Undaria’.

3. Figure 4b: was a bit blurry.

4. Revisions for the references were required. For examples, the issue numbers of references [1], [5] and [6] were not necessary, the pages for [2] could be updated as ‘2022, 30, 785-809.’, and the journal names ‘Carbohydrate polymers’ in [2], ‘Cell research’ in [9], ‘Marine drugs’ in [14] were not recorded as usual.

Author Response

RESPONSE TO REVIEWER 2

Comment-1: Biological properties and health-promoting functions of Laminarin: A comprehensive review of preclinical and clinical studies by Shanmugapriya Karuppusamy et al described the diverse pharmacological properties of laminarin, such as antioxidant, immunomodulatory and antiobesity. Moreover, mechanisms of action were summarized detailedly in this review. This is a well-organized comprehensive review, providing an overview of recent advances, and identifies gaps and opportunities for further research in this field.

Response:  Thank you for these overall positive comments.

Only minor revisions as following:

Comment-2: The structure in Figure 2 was too large.

Response:  Thank you. Figure 2 has been modified and is included in line number, 94 of the revised manuscript. 

Comment-3: P3L97: Italics for the species and genera names for ‘Laminaria digitata, Saccharina latissima, Laminaria japonica, Ecklonia kurome, Eisenia bicyclis, Ascophyllum, Fucus, Undaria’.

Laminarin has been extracted from different brown macroalgal species including Laminaria digitata, Saccharina latissima, Laminaria japonica, Ecklonia kurome, Eisenia bicyclis and to lesser extents in Ascophyllum, Fucus, and Undaria species located in Asian and a few European countries

Response:  Thank you for your suggestion. Text has been modified and included in the revised manuscript.

Sentences are included in line number 96 as follows:

“~Laminarin has been extracted from different brown macroalgal species including Laminaria digitata, Saccharina latissima, Laminaria japonica, Ecklonia kurome, Eisenia bicyclis and to lesser extents in Ascophyllum, Fucus, and Undaria species obtained from Asian and European countries~” 

Comment-4: Figure 4b: was a bit blurry.

Response:  Thank you. Figure has been modified and included in line number 399 of the revised manuscript. 

Comment-5: Revisions for the references were required. For examples, the issue numbers of references [1], [5] and [6] were not necessary, the pages for [2] could be updated as ‘2022, 30, 785-809.’, and the journal names ‘Carbohydrate polymers’ in [2], ‘Cell research’ in [9], ‘Marine drugs’ in [14] were not recorded as usual.

Response:  Thank you for your suggestion. Text has been modified and included in reference part of in the revised manuscript.

Reviewer 3 Report

The manuscript is well structured, however, there is an keystone which need to be more developed:

- Extraction, purification and characterization to the normal reader understand the major differences between laminarin and other polysaccharides from brown seaweeds.

Other important task is in Table 2 and 3 -  need to be reworked to have more information in the studies referenced and not induce questions about how is laminarin extracted or it was buyed. Moreover, it was characterized because that is an importante step towards the bullet point of this manuscript.

Other minor question are in the annex

Author Response

RESPONSE TO REVIEWER 3

The manuscript is well structured, however, there is an keystone which need to be more developed:

Comment-1: Extraction, purification and characterization to the normal reader understand the major differences between laminarin and other polysaccharides from brown seaweeds.

Other important task is in Table 2 and 3 -  need to be reworked to have more information in the studies referenced and not induce questions about how is laminarin extracted or it was buyed. Moreover, it was characterized because that is an important step towards the bullet point of this manuscript.

Response:  Thank you for your suggestion. Text has been modified in the revised manuscript. We modified and included extraction methods in Table 2 (line number 178-197) and Table 3 (line number 314-323). 

Comment-2: Other minor question are in the annex

Response:  Thank you for your suggestion. Text has been modified and included in the revised manuscript.

Reviewer 4 Report

The author summarizes laminarin's biological activities, including mechanisms of action, effects on human health, and reported health benefits. The review is interesting; therefore, the author should improve their content. Including:

  • Figure 2 is too large; please resize it to a smaller size.
  • The species should be italicized (ex. L96–99; L137). Please double-check everything.
  • Please correct the entire Table 1 (specific species, specific functions, and specific references).
  • Please include the full name before abbreviating it (ex. OC, OT, etc.).
  • Figure 4-8's resolution should be improved.
  • Please limit the number of times a figure is ‘reprinted’. Did you truly obtain permission? Please also attach the evidence.
  • Please include a section on the extraction and purification of laminarin, as well as the advanced methods for detecting and characterizing laminarin (mass spectrometry, NMR, FTIR, etc.).
  • Please include the laminarin yield of each species (maybe in a table); could laminarin be synthesized?
  • It would be nice to present a graph comparing laminarin, fucoidan, ulvan, carrageenan, and alginate.
  • Please include the number of clinical trials and studies in the table.

Author Response

RESPONSE TO REVIEWER 4

The author summarizes laminarin's biological activities, including mechanisms of action, effects on human health, and reported health benefits. The review is interesting; therefore, the author should improve their content. Including:

Comment-1: Figure 2 is too large; please resize it to a smaller size.

Response:  Thank you for your suggestion. Figure has been modified and included in line number 94 of the revised manuscript. 

Comment-2: The species should be italicized (ex. L96–99; L137). Please double-check everything.

Response:  Thank you for your suggestion. Text has been modified and included in the revised manuscript.

Sentences are included in line number 97-98; 140 as follows:

“~Laminarin has been extracted from different brown macroalgal species including Laminaria digitata, Saccharina latissima, Laminaria japonica, Ecklonia kurome, Eisenia bicyclis and to lesser extents in Ascophyllum, Fucus, and Undaria species obtained from Asian and European countries ~”

“~Garcia-Vaquero et al. [23] investigated the antioxidant activity in laminarin from Laminaria digitata using DPPH and FRAP methods. The antioxidant and antimicrobial activities of crude laminarin extract were also examined by Kadam et al. [40] who confirmed….~” 

Comment-3: Please correct the entire Table 1 (specific species, specific functions, and specific references).

Response:  Thank you for your suggestion. Table has been modified and included in line number 158 of the revised manuscript. 

Comment-4: Please include the full name before abbreviating it (ex. OC, OT, etc.).

Response:  Thank you for your suggestion. Changes now included in the revised manuscript.

Comment-5: Figure 4-8's resolution should be improved.

Response:  Thank you for your suggestion. Figures have been modified and included in line number 382, 399, 424, 464, 496, 500 of the revised manuscript. In addition, figure 1 was modified in line number 65. 

Comment-6: Please limit the number of times a figure is ‘reprinted’. Did you truly obtain permission? Please also attach the evidence.

Response:  Thank you for your suggestion. The copyright permission where required for Figures 3/4/5/6/7/8 are enclosed as attachment. Figures 1 and 2 (line number 65, 94) were prepared by the authors.

Comment-7: Please include a section on the extraction and purification of laminarin, as well as the advanced methods for detecting and characterizing laminarin (mass spectrometry, NMR, FTIR, etc.).

Response:  Thank you for your suggestion. Paper has been modified, now included as one column in Table 2 (line number 178-197) and Table 3 (line number 314-323) of the revised manuscript.

Comment-8: Please include the laminarin yield of each species (maybe in a table); could laminarin be synthesized?

Response:  Thank you for your suggestion. Text has been modified and included in Table 1 (line number 158-162), Table 2 (line number 178-197) and Table 3 (line number 314-323) of the revised manuscript.

Comment-9: It would be nice to present a graph comparing laminarin, fucoidan, ulvan, carrageenan, and alginate.

Response:  Thank you for your suggestion. Text has been modified and included in table 1 (line number 158-162). As other reviewers requested, we considered to include all information in one table as table 1 (line number 158-162) of the revised manuscript.

Comment-10: Please include the number of clinical trials and studies in the table.

Response:  Thank you for your suggestion. This information is now included in Table 3 (line number 314-323)  of the revised manuscript.

Round 2

Reviewer 1 Report

The manuscript can be accepted

Author Response

Comment-1: The manuscript is well structured, however, there is more informatione which need to be more developed:

Major important task is to discuss and critical analysis about the Table 2 and 3, with information added, what the author think about the data? Because, there are several studies that are not done in purified Laminarin but with brown polymers extract, is laminarin effect or can be other compound effect. How authors can be so secure that is only laminarin? The manuscript needs to be discussed with the literature cited, in a general point of view.

Response:  Thank you for your suggestion. This information is now included in Table 2 and 3 of the revised manuscript. The laminarin has been extracted using different methods from several brown seaweed species. Therefore, several studies demonstrated that laminarin or purified Laminarin or crude extarcts or fraction provides a wide variety of biological activities. These information are mentioned with the the literature cited in the manuscript, line number 138, 144 and so on. We included this information in Table 2, line number 175 and Table 3, 301. We modified and included in 2 & 6 columns of Table 2 (line number 176-189) and Table 3 (line number 313-315).

Comment-2: Other important task related to the above task is: - Extraction, purification and characterization need to be added to be clear all the message, because the authors focus in Laminarin (which is very good), however, was demonstrated there is other polymer in brown seaweed, that can difficult the task of extracting and purify Laminarin of the algal biomass.

Response:  Thank you for your suggestion. This information is now included in Table 2 and 3 of the revised manuscript. In table 2 and 3, under heading-Extraction, purification and characterization in second coluimn, laminarin or purified Laminarin or crude extarcts or fraction has been modified and included in the revised manuscript.

Response:  Thank you for your suggestion. This information is modified and included in Table 1 (line number 79), 2 (line number 176) and 3 (line number 313) of the revised manuscript.

Reviewer 3 Report

Thank you for the revision in the manuscript, although, there are several comments that need to be developed, see below and in the annex:

The manuscript is well structured, however, there is more informatione which need to be more developed:

Major important task is to discuss and critical analysis about the Table 2 and 3, with information added, what the author think about the data? Because, there are several studies that are not done in purified Laminarin but with brown polymers extract, is laminarin effect or can be other compound effect. How authors can be so secure that is only laminarin? The manuscript needs to be discussed with the literature cited, in a general point of view.

Other important task related to the above task is: - Extraction, purification and characterization need to be added to be clear all the message, because the authors focus in Laminarin (which is very good), however, was demonstrated there is other polymer in brown seaweed, that can difficult the task of extracting and purify Laminarin of the algal biomass.

Author Response

(The authors gave the same response as above.)

Reviewer 4 Report

Many changes were made by authors, and efforts to improve were appreciated. Some issues, however, must be corrected.
Please correct the following information.

1) Table 2. Summary of preclinical studies with respect to biological activity and health benefits of laminarin (n=20) and Table 3. Summary of clinical studies with respect to biological activity and health benefits of laminarin (n=7).

They should not be written as "laminarin" because they were an extract or fraction. The extract contained several components, not just laminarin.

Table 3 focuses on extraction, purification, yield, and characterisation, but it also includes information on:

   a) The trial product (CM3, http://www.buyephed-radietpills.com/cm3-
alginate.html) is a patented formulation that contains a compressed, lyophilized sodium–alginate active complex, based on brown seaweed, L. digitata.

  b) A group of 30 patients who were dilated pre-operatively with Laminaria tents.

Author Response

Comment-1: Many changes were made by authors, and efforts to improve were appreciated. Some issues, however, must be corrected.

Please correct the following information.

1) Table 2. Summary of preclinical studies with respect to biological activity and health benefits of laminarin (n=20) and Table 3. Summary of clinical studies with respect to biological activity and health benefits of laminarin (n=7).

They should not be written as "laminarin" because they were an extract or fraction. The extract contained several components, not just laminarin.

Table 3 focuses on extraction, purification, yield, and characterisation, but it also includes information on:

  1. a) The trial product (CM3, http://www.buyephed-radietpills.com/cm3-alginate.html) is a patented formulation that contains a compressed, lyophilized sodium–alginate active complex, based on brown seaweed, L. digitata.
  2. b) A group of 30 patients who were dilated pre-operatively with Laminaria tents.

Response:  Thank you for your suggestion. This information is now included in Table 2 & 3 (line number 176-189, 313-315)  of the revised manuscript.

Sentences are included as follows:

“~Table 2. Summary of preclinical studies with respect to biological activity and health ben-efits of laminarin extract or fraction (n=17)

Table 3. Summary of clinical studies with respect to biological activity and health benefits of laminarin extract or fraction and related products (n=4). ~” 

Round 3

Reviewer 3 Report

The authors was answered all of my questions.